# Steroid profiling in human primary teeth via liquid chromatography-tandem mass spectrometry for long-term retrospective steroid measurement

Ruolan S. Wu[1,2]*, Jordan E. Hamden[1,3], Melody Salehzadeh[1,3], Michael X. Li[1,2], Asmita Poudel[1,2], Kim L. Schmidt[4], Michael S. Kobor[4,5], Kiran K. Soma[1,2,3]

1 Djavad Mowafaghian Centre for Brain Health, University of British Columbia, Vancouver, BC, Canada, 2 Department of Psychology, University of British Columbia, Vancouver, BC, Canada, 3 Department of Zoology, University of British Columbia, Vancouver, BC, Canada, 4 Edwin S.H. Leong Centre for Healthy Aging, University of British Columbia, Vancouver, BC, Canada, 5 Department of Medical Genetics, University of British Columbia, Vancouver, BC, Canada

* ruolan.s.wu@gmail.com

**Data Availability Statement:** All data files are available from the OSF database (accession number DOI 10.17605/OSF.IO/RQT65).

## Abstract

Steroid hormones are important modulators of many physiological processes, and measurements of steroids in blood, saliva, and urine matrices are widely used to assess endocrine pathologies and stress. However, these matrices cannot be used to retrospectively assess early-life stress and developmental endocrine pathologies, because they do not integrate steroid levels over the long term. A novel biological matrix in which to measure steroids is primary teeth (or "baby teeth"). Primary teeth develop early in life and accumulate various endogenous molecules during their gradual formation. Here, we developed and validated the first assay to measure steroids in human primary teeth using liquid chromatography-tandem spectrometry (LC-MS/MS). Our assay is highly sensitive, specific, accurate, and precise. It allows for the simultaneous quantification of 17 steroids in primary teeth (16 of which have not been examined previously in primary teeth). Overall, steroid levels in primary teeth were relatively low, and 8 steroids were quantifiable. Levels of dehydroepiandrosterone, cortisol, and progesterone were the highest of the 17 steroids examined. Next, we used this assay to perform steroid profiling in primary teeth from males and females. The same 8 steroids were quantifiable, and no sex differences were found. Levels of androgens (androstenedione and testosterone) were positively correlated, and levels of glucocorticoids (cortisol, cortisone, corticosterone, 11-dehydrocorticosterone) were also positively correlated. These data demonstrate that multiple steroids can be quantified by LC-MS/MS in human primary teeth, and this method potentially provides a powerful new way to retrospectively assess early-life stress and developmental endocrine pathologies.

**Funding:** Natural Sciences and Engineering Research Council of Canada (NSERC) Discovery Grant (RGPIN-2019-04837) and Discovery Accelerator Supplement (RGDAS-2019-00033) to KKS; UBC Grants for Catalysing Research Clusters (F18-05858), and a New Frontiers in Research Fund Grant (F21-04739) to MSK; Michael Smith Health Research BC and CLEAR Foundation Postdoctoral Fellowship to JEH; and NSERC CGS-M and CGS-D fellowships to MS. The funders had no role in study design, data collection and analysis, decision to publish, or preparation of the manuscript.

**Competing interests:** The authors have declared that no competing interests exist.

## Introduction

Steroid hormones are important modulators of a myriad of physiological processes, including metabolism, mood and cognition, and reproduction [1–4]. Steroid hormones are secreted by the adrenal glands and gonads, and regulated by the hypothalamic-pituitary-adrenal (HPA) and hypothalamic-pituitary-gonadal (HPG) axes [4]. Dysregulation of steroid hormone secretion, for example, by chronic stress or endocrine disrupting chemicals (EDCs), can lead to various diseases, such metabolic syndrome, mood disorders, and infertility [5,6].

Steroid hormones are used as biomarkers because they can reflect physiological and pathological states [7–9]. A common approach to quantify stress is measuring cortisol in saliva, blood, or urine [10–15]. However, these biological matrices do not allow for long-term retrospective steroid analysis [16–18]. Hair cortisol shows promise for retrospective stress assessment, but only captures the previous 6 months [19–22]. Furthermore, hair cortisol is difficult to measure in bald or thin-haired individuals and is affected by bleaching and other factors [23,24].

Primary (deciduous or baby) teeth possess unique properties that allow for retrospective steroid hormone analysis extending back to the prenatal period, which is particularly challenging to capture with other biological matrices [25]. Saliva, blood, urine, or hair cannot be sampled in the fetus, but primary teeth retains a permanent record of their incremental formation from the second trimester of pregnancy to 18–36 months after birth [25–31]. Exposure to stress during this developmental period is reflected as alterations in primary teeth structure [25,26,32]. For example, children exposed to perinatal stress have more pronounced demarcations on their primary teeth, known as stress lines [33–35]. Environmental exposures are also reflected in primary teeth's chemical composition [27,28,35,36]. For example, infants who live in areas with higher ambient heavy metals have higher levels of heavy metals in their primary teeth [37–39].

Measuring steroids in primary teeth may be a useful and novel way to quantitively assess early-life stress (ELS), as well as other developmental and endocrine disorders. Various organic and inorganic compounds have been detected in primary teeth and are deposited during tooth formation, which begins *in utero* [27,28,35–40]. Thus, dysregulations of steroid hormone secretion during perinatal life may also be reflected in dental steroid levels. To our knowledge, only two groups have measured dental steroids, both using an enzyme-linked immunoassay (ELISA) designed for salivary cortisol [40–42]. This ELISA was not validated for use in teeth, and no other steroids were investigated.

Here, we developed and validated a protocol for measuring multiple dental steroids at low concentrations via liquid chromatography tandem mass spectrometry (LC-MS/MS). Tooth surfaces and pulp chambers were thoroughly cleaned. Then, teeth were pulverized, steroids were extracted via liquid-liquid extraction, and 17 steroids were measured by LC-MS/MS. The assay is highly sensitive, specific, accurate, and precise, and matrix effects are minimal or absent. After assay development, a panel of 17 steroids was measured in primary teeth from males and females.

## Materials and methods

### Subjects

All protocols were approved by the University of British Columbia (UBC) Children's and Women's Research Ethics Board (H19-00444). Participants were recruited between Aug 20, 2021 –June 21, 2022. Informed consent was obtained from parents via written consent forms.

**Table 1. Characteristics of human primary tooth samples used to study sex difference.**

|  | Male | Female | *p* |
|---|---|---|---|
| Age at tooth loss (yr) | 5.3 ± 0.2 | 6.2 ± 0.3 | 0.04 |
| Time from tooth loss to assay (yr) | 2.9 ± 1.0 | 1.0 ± 0.1 | 0.04 |
| Tooth mass (mg) | 75.3 ± 6.3 | 66.5 ± 4.5 | 0.27 |

Note: Data are shown as mean ± *SEM*. *n* = 8 males and 9 females. Significance criterion was set at $p \leq 0.05$.

**Development and validation.** Human primary teeth were from male and female participants ranging from 4 to 8 years of age. Parents from British Columbia, Canada were recruited using social media advertisements. Participants mailed the tooth and questionnaire to the BC Children's Hospital using a study kit provided.

A total of 23 primary teeth of all types (incisors, canines, molars) were used for assay development. All primary teeth were naturally-shed and had no decay or fillings. A subset of teeth was used for method validations of parallelism (n = 3) and recovery (n = 6). The remaining 14 teeth were used to assess general steroid concentrations in primary teeth.

**Sex differences in dental steroid levels.** Human primary teeth that were naturally shed were used from male and female participants ages 4 to 8 years from B.C. Canada (n = 8 males and 9 females). Sex was obtained via parental reports and validated using PCR. All teeth were confirmed by a dentist to be bottom central incisors and free of anomalies, decay, or fillings (Table 1).

## Tooth sample preparation

Teeth were stored at room temperature prior to sampling. In previous studies of teeth, feathers, hair, nails, and baleen, steroids were stable at room temperature [40–49]. Tooth surfaces were thoroughly scrubbed with a toothbrush and deionized water to remove any saliva, blood, or contaminants. Teeth were split in half with a MicroCryoCrusher (Biospec, Bartlesville, OK) to expose the pulp chambers. Each tooth was sonicated in phosphate buffered saline (Sigma-Aldrich, Oakville, ON) for 15 min at room temperature to remove pulp. Any residual pulp or blood was carefully removed by scrubbing with a toothbrush and deionized water. Next, each tooth was crushed to fine powder using the MicroCryoCrusher and the powder was weighed. The MicroCryoCrusher and toothbrush were cleaned thoroughly with 70% ethanol between samples.

Cells in pulp/blood (obtained after sonication) were genotyped by the UBC Genotyping Facility. PCR was used to detect the presence of the male-specific SRY gene, located on the Y chromosome. Accuracy of the assay was confirmed by genotyping tooth samples from subjects of known sex.

## Steroid extraction

Steroids were extracted as previously described [50]. The powder from each tooth was placed in a 2 ml polypropylene tube (Sarstedt, Montreal, QC) containing 5 zirconium ceramic oxide beads (1.4 mm diameter) (Fisher Scientific, Ottawa, ON). Proteins were precipitated and steroids were extracted by adding 1 ml HPLC-grade acetonitrile to each tube and homogenizing using a bead mill homogenizer for 30 s at 4 m/s (Omni International Inc., Kennesaw, GA). Samples were centrifuged for 5 min at 16,100× *g*, and 1 ml of the supernatant was transferred to 12x75 mm borosilicate glass culture tubes (pre-cleaned with HPLC-grade methanol) (Fisher Scientific, Ottawa, ON). Next, 500 µl HPLC-grade hexane was added to each culture tube and

vortexed briefly. Samples were centrifuged for 2 min at 3,200× *g*, and hexane was removed and discarded. Samples were dried in a vacuum centrifuge at 60°C for 45 min (ThermoElectron, Waltham, MA). Dried residues were resuspended in 55 µl of 1:3 HPLC-grade methanol:MilliQ water, vortexed briefly, and centrifuged for 1 min at 3,200× *g*. The supernatant was then transferred to a 0.6 ml polypropylene microcentrifuge tube and centrifuged for 2 min at 16,100× *g* to pellet any debris. The entire supernatant was transferred to a 2 ml screw top vial (Agilent, Santa Clara, CA) fitted with a glass insert and stored at -20°C.

## Steroid analysis by LC-MS/MS

Steroid analysis by LC-MS/MS was conducted as previously described [50]. Briefly, samples were loaded into an autosampler maintained at 15°C, and 45 µl of each sample was injected into a Nexera X2 UHPLC system (Shimadzu Corp., Japan), passed through an in-line filter, Security-Guard™ ULTRA C18 UHPLC guard column (2.1 mm) (Phenomenex) and a Kinetex® Core-shell C18 column (2.1 x 50 mm; 2.6 µm; at 40°C) (Phenomenex) with gradient binary mobile system using 0.1 mM ammonium fluoride in MilliQ water as mobile phase A and HPLC-grade methanol as mobile phase B (MPB). The flow rate was 0.4 ml/min. During loading, MPB was at 10% for 0.5 min, and then the gradient profile began at 42% MPB for 3.5 min before being ramped to 60% MPB until 9.4 min. From 9.4 to 9.5 min the gradient was 60%–70% MPB, then it was ramped to 98% MPB until 11.9 min, and finally a column wash at 98% MPB until 13.4 min. The MPB was then returned to starting conditions for 1 min. Total run time was 14.9 min. The injection needle was rinsed before and after each sample injection with 100% methanol.

The quantification of steroids was performed on AB Sciex 6500 Qtrap triple quadrupole tandem mass spectrometer (AB Sciex LLC, MA). Steroids were quantified with scheduled multiple reaction monitoring (sMRM) with two mass transitions for each analyte (dehydroepiandrosterone (DHEA), androstenedione, 5α-dihydrotestosterone, testosterone, 11-deoxycortisol, cortisol, cortisone, 11-deoxycorticosterone, corticosterone, 11-dehydrocorticosterone (DHC), progesterone, pregnenolone, allopregnanolone, estrone, 17β-estradiol, estriol, and aldosterone) and one mass transition for each internal standard. Electrospray ionization with negative mode was used for estrone, 17β-estradiol, and estriol whereas positive mode was used for all other steroids. Source and compound dependent parameters were optimized by direct infusion of pure standards. Calibration curves, blanks, double blanks, and quality controls were processed alongside samples. Standards ranged from 0.005 to 100 pg/µl. 10 µl of each standard was used to make the calibration curves. The calibration curves were then processed alongside the samples under the same extraction procedure. Both the calibration curves and the teeth samples were dried in a vacuum centrifuge and resuspended in the same final volume of 55 µl of 1:3 HPLC-grade methanol:MilliQ water. The calibration curve range was 0.05 to 1000 pg per tube (0.91 to 18,182 pg/ml) for all steroids (excluding DHEA, pregnenolone, and allopregnanolone) and prepared in 50:50 HPLC-grade methanol:MilliQ water. The calibration curve range was 2 to 1000 pg per tube (36.4 to 18,182 pg/ml) for DHEA, pregnenolone, and allopregnanolone. Accuracy and precision were assessed by measuring 2 pg and 200 pg quality controls in triplicate in each assay.

## Validations

Matrix effects were determined by comparing the internal standards' response in teeth matrix versus their response in neat solution. Following Eq (1) was used to calculate matrix effect.

$$Matrix\ Effect = \frac{Response\ of\ internal\ standard\ in\ teeth\ matrix}{Response\ of\ internal\ standard\ in\ neat\ solution} * 100 \qquad (1)$$

Prior to homogenization and extraction, 50 μl of deuterated internal standards (dehydro-epiandrosterone-d6, testosterone-d5, cortisol-d4, corticosterone-d8, progesterone-d9, pregnenolone-d4, allopregnanolone-d4, 17β-estradiol-d4, and aldosterone-d7 C/D/N Isotopes Inc., Pointe-Claire, Canada) in 50:50 HPLC-grade methanol:MilliQ water were added to the calibration curves, quality controls, blanks, and samples (40 pg/sample of DHEA-d6, pregnenolone-d4, allopregnanolone-d4, and 20 pg/sample of all other internals standards). DHEA-d6 was used as an internal standard for DHEA; testosterone-d5 was used for androstenedione, 5α-dihydrotestosterone, and testosterone; cortisol-d4 was used for cortisol and cortisone; corticosterone-d8 was used for 11-deoxycortisol, 11-deoxycorticosterone, corticosterone, and DHC; progesterone-d9 was used for progesterone; pregnenolone-d4 was used for pregnenolone; allopregnanolone-d4 was used for allopregnanolone; 17β-estradiol-d4 was used for estrone, 17β-estradiol, and estriol; and aldosterone-d7 was used for aldosterone. Furthermore, to assess matrix effects, we determined whether serial-diluted tooth samples showed linearity and parallelism with the calibration curve. Teeth were homogenized as described above. Homogenates from 3 teeth were pooled in a 20 ml glass scintillation vial (Fisher Scientific, Ottawa, ON) pre-cleaned with HPLC-grade methanol. As dental steroid concentrations were low, the pooled homogenate was serially diluted by 2-fold, from 1 to 1/128. Steroids were extracted as described above. Parallelism was assessed by comparing the slopes of the serial-diluted teeth and the calibration curves.

To assess the recovery of exogenous steroids after extraction, homogenates from 6 teeth were pooled in a pre-cleaned 20 ml glass scintillation vial, then aliquoted into 2 ml polypropylene tubes. The tubes were split into two groups (n = 6/group) and spiked with either known amounts of steroid (based on expected steroid concentrations; 50 pg of DHEA, 5 pg of cortisol, and 2 pg of all other steroids) or unspiked. Steroids were extracted as described above. To assess recovery, unspiked homogenate values were subtracted from spiked homogenate values, and the difference was divided by the amount of steroid added.

## Data analysis

Statistics were conducted using GraphPad Prism version 9.4.1 (GraphPad Software). When necessary, data were log transformed before analysis. Sex differences in steroid levels were analyzed by Student's t-tests, or in the case of unequal variances, by Welch's t-tests. The linearity of calibration curves was analyzed by Pearson correlations. The correlations between steroid levels were examined using Spearman's rho correlations. Significance criterion was set at $p \leq 0.05$. Graphs show the mean ± *SEM* and are presented using the non-transformed data.

## Results

### Assay development and validation

Steroids were analyzed by LC-MS/MS using scheduled multiple reaction monitoring with two mass transitions for each analyte. The use of two unique mass transitions and a unique retention time for each analyte provided excellent specificity (Table 2) (Fig 1).

The calibration curve consisted of 13 points ranging from 0.05 to 1000 pg per tube (0.91 to 18,182 pg/ml) for androstenedione, 5α-dihydrotestosterone, testosterone, 11-deoxycortisol, cortisol, cortisone, 11-deoxycorticosterone, corticosterone, DHC, progesterone, estrone, 17β-estradiol, estriol and aldosterone. The calibration curve ranged from 2 to 1000 pg per tube (36.4 to 18,182 pg/ml) for DHEA, pregnenolone, and allopregnanolone. The on-column amount ranged from 0.041 pg to 818.2 pg per injection for all steroids (except DHEA, pregnenolone, and allopregnanolone). The on-column amount ranged from 1.64 pg to 818.2 pg per injection for DHEA, pregnenolone, and allopregnanolone. All steroids showed excellent

**Table 2. Scheduled multiple reaction monitoring for LC-MS/MS.**

| Steroid | Ion Mode | Retention Time (min) | Quantifier m/z | Qualifier m/z |
|---|---|---|---|---|
| DHEA | ESI + | 8.76 | 271.1→253.0 | 271.1→213.2 |
| DHEA-d6 | ESI + | 8.71 | 277.1→219.2 | - |
| Androstenedione | ESI + | 7.34 | 287.2→97.2 | 287.2→109.1 |
| 5α-Dihydrotestosterone | ESI + | 9.69 | 291.2→255.3 | 291.2→159.1 |
| Testosterone | ESI + | 8.12 | 289.0→97.0 | 289.0→109.1 |
| Testosterone-d5 | ESI + | 8.05 | 294.0→100.0 | - |
| Cortisol | ESI + | 3.66 | 363.3→121.2 | 363.3→327.1 |
| Cortisone | ESI + | 3.30 | 361.0→163.0 | 361.0→121.1 |
| Cortisol-d4 | ESI + | 3.63 | 367.2→121.1 | - |
| 11-Deoxycortisol | ESI + | 6.21 | 347.0→109.0 | 347.0→97.0 |
| 11-Deoxycorticosterone | ESI + | 7.94 | 331.0→97.0 | 331.0→109.1 |
| Corticosterone | ESI + | 5.59 | 347.1→121.1 | 347.1→91.1 |
| DHC | ESI + | 3.99 | 345.0→121.0 | 345.0→301.0 |
| Corticosterone-d8 | ESI + | 5.45 | 355.1→124.9 | - |
| Progesterone | ESI + | 10.37 | 315.2→97.0 | 315.2→109.1 |
| Progesterone-d9 | ESI + | 10.33 | 324.2→100.0 | - |
| Pregnenolone | ESI + | 10.64 | 299.1→159.1 | 299.1→105.1 |
| Pregnenolone-d4 | ESI + | 10.62 | 303.0→159.1 | - |
| Allopregnanolone | ESI + | 11.12 | 301.1→283.2 | 301.1→135.0 |
| Allopregnanolone-d4 | ESI + | 11.10 | 305.0→114.8 | - |
| Estrone | ESI - | 7.58 | 269.0→145.0 | 269.0→143.0 |
| 17β-Estradiol | ESI - | 7.63 | 271.0→145.0 | 271.0→143.0 |
| Estriol | ESI - | 2.70 | 287.1→171.0 | 287.1→144.9 |
| 17β-Estradiol-d4 | ESI - | 7.59 | 275.0→147.0 | - |
| Aldosterone | ESI + | 2.67 | 361.3→315.4 | 361.3→343.3 |
| Aldosterone-d7 | ESI + | 2.63 | 368.2→322.3 | - |

Abbreviations: LC-MS/MS, liquid chromatography tandem mass spectrometry; DHEA, dehydroepiandrosterone; DHC, 11-Dehydrocorticosterone; ESI, electrospray ionization.

linearity and displayed a coefficient of determination ($r^2$) greater than 0.99 and p-values less than 0.0001 (Table 3) (Fig 2). The injection volume was 45 μl for each sample.

Blanks, double blanks (without internal standards), and quality controls were processed alongside samples. The signal to noise ratios for blanks and double blanks were below 3, as expected. The blank sample also exhibited no interference with the analyte' transitions. All quality controls were within 5% of their expected concentration, and thus accuracy was within the acceptable limit of 20% of the expected concentrations. Intra-assay and inter-assay variation (% CV) were below 5%, confirming the precision of the assay (Table 3).

Matrix effects were assessed by adding internal standards (DHEA-d6, testosterone-d5, cortisol-d4, corticosterone-d8, progesterone-d9, pregnenolone-d4, allopregnanolone-d4, 17β-estradiol-d4, and aldosterone-d7) to all samples (except the double blanks). The internal standard peak areas of the tooth samples were 100–110% of the internal standard peak areas of the calibration curve samples (without matrix), indicating the absence of matrix effects. The matrix effect was also evaluated by assessing the parallelism of calibration curves with serially diluted teeth samples. To assess parallelism, teeth (n = 3) were pooled and serial diluted. The

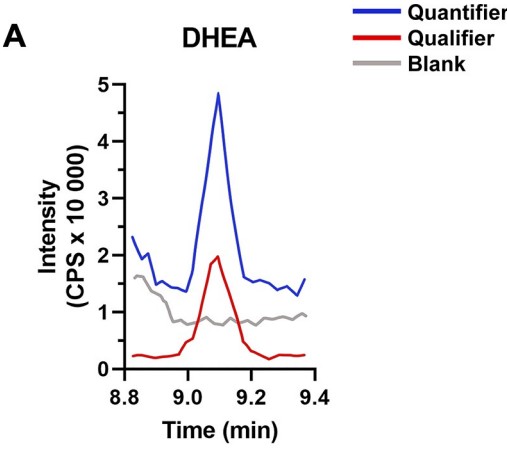

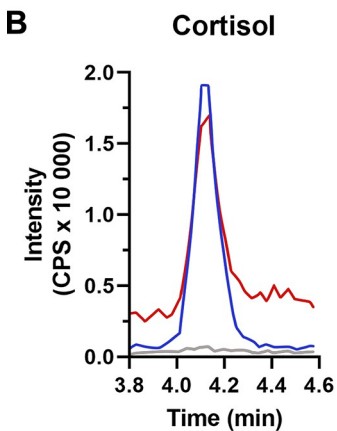

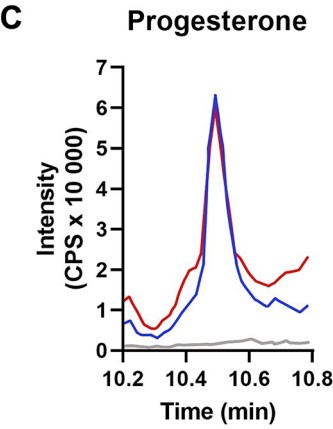

**Fig 1. Representative chromatograms of the 3 most abundant steroids in human primary teeth samples as measured by liquid chromatography tandem mass spectrometry (LC-MS/MS). (A)** dehydroepiandrosterone (DHEA), **(B)** cortisol, and **(C)** progesterone. Quantifier transition (blue) and qualifier transition (red) are shown over the water blank (grey), and all analytes had unique transitions. Retention time is shown on the x-axis, and all analytes had unique retention times. Blank shows no interference with the transitions of steroids demonstrating methods' selectivity for the target steroids. Chromatograms shown were from primary tooth samples. CPS, counts per second.

**Table 3. Coefficient of determination ($r^2$), p-value and lower limit of quantification (LLOQ) of LC-MS/MS calibration curves.**

| Steroid | $r^2$ | $p$ | LLOQ (pg/10 µl) | Intra-assay %CV | Inter-assay %CV |
|---|---|---|---|---|---|
| DHEA | 0.9946 | <0.0001 | 2.0 | 1.0 | 4.8 |
| Androstenedione | 0.9912 | <0.0001 | 0.05 | 2.4 | 3.2 |
| 5α-Dihydrotestosterone | 0.9967 | <0.0001 | 0.05 | 0.3 | 4.7 |
| Testosterone | 0.9991 | <0.0001 | 0.05 | 0.3 | 5.1 |
| Cortisone | 0.9959 | <0.0001 | 0.05 | 1.1 | 3.1 |
| Cortisol | 0.9976 | <0.0001 | 0.05 | 5.8 | 6.9 |
| 11-Deoxycortisol | 0.9969 | <0.0001 | 0.05 | 0.7 | 3.4 |
| 11-Deoxycorticosterone | 0.9982 | <0.0001 | 0.05 | 0.6 | 5.4 |
| DHC | 0.9962 | <0.0001 | 0.05 | 3.9 | 4.5 |
| Corticosterone | 0.9993 | <0.0001 | 0.05 | 2.1 | 4.2 |
| Progesterone | 0.9975 | <0.0001 | 0.05 | 2.0 | 0.3 |
| Pregnenolone | 0.9987 | <0.0001 | 2.0 | 5.9 | 7.2 |
| Allopregnanolone | 0.9906 | <0.0001 | 2.0 | 3.7 | 4.6 |
| Estrone | 0.9966 | <0.0001 | 0.05 | 3.0 | 5.4 |
| Estriol | 0.9988 | <0.0001 | 0.05 | 5.0 | 5.5 |
| 17β-estradiol | 0.9978 | <0.0001 | 0.05 | 1.3 | 4.4 |
| Aldosterone | 0.9980 | <0.0001 | 0.05 | 2.8 | 3.1 |

Note: The limit of detection (LOD) for aldosterone, DHC, cortisol, and corticosterone was 0.025 pg/10 µl. The LODs for progesterone, pregnenolone, and DHEA were 0.01 pg/10 µl, 0.015 pg/10 µl, and 0.2 pg/10 µl, respectively. The LOD was selected as the lowest concentration at which the analyte produced a signal-to-noise (S/N) ratio ≥ 3. The LLOQ was selected as the lowest concentration with a S/N ratio ≥ 5, with an accuracy within ±20% of the nominal value and a coefficient of variation (CV) within ±20% [51].

slopes for serial-diluted teeth and calibration curves were similar for all steroids (differences in slopes were 0.3–16.2%) (Table 4).

Finally, to assess recovery, teeth were pooled (n = 6) and split into two groups, spiked with either known amounts of steroid or vehicle. Percent recovery was calculated by subtracting the quantity of steroids in the vehicle tissue pools from those of the spiked pools and dividing by the amount of steroid added. Recovery was 95–105% of the expected value for all steroids (Table 5).

In a pilot study, we measured 17 steroids in primary teeth ($n$ = 14) to determine approximate steroid levels. Eight steroids were above the lower limit of quantification (Fig 3), and DHEA, cortisol, and progesterone were the most abundant (Fig 3A and 3C). Cortisone, corticosterone, DHC, androstenedione, and testosterone were also quantifiable (Fig 3B and 3D). 5α-dihydrotestosterone, 11-deoxycortisol, 11-deoxycorticosterone, pregnenolone, allopregnanolone, estrone, 17β-estradiol, estriol, and aldosterone were non-detectable in all samples.

## Sex differences

17 primary bottom central incisors were available to us ($n$ = 8 males and 9 females) for the examination of sex differences. Consistent with the pilot study, we were able to quantify the same 8 steroids: DHEA, cortisol, progesterone, cortisone, corticosterone, DHC, androstenedione, and testosterone. Again, DHEA, cortisol, and progesterone were the most abundant steroids. No significant differences were found between males and females (all $p$ values > 0.05 in all cases) (Fig 4). As in the pilot study, the other 9 steroids (5α-dihydrotestosterone, 11-deoxycortisol, 11-deoxycorticosterone, pregnenolone, allopregnanolone, estrone, 17β-estradiol, estriol, and aldosterone) were non-detectable in all samples. As

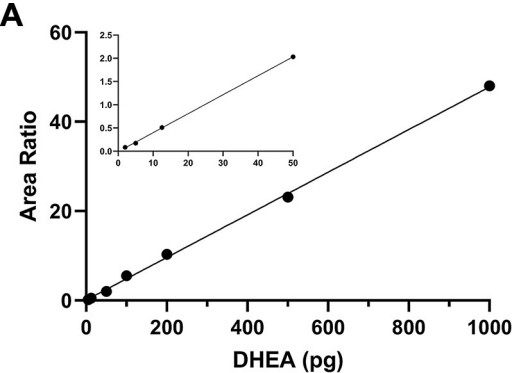

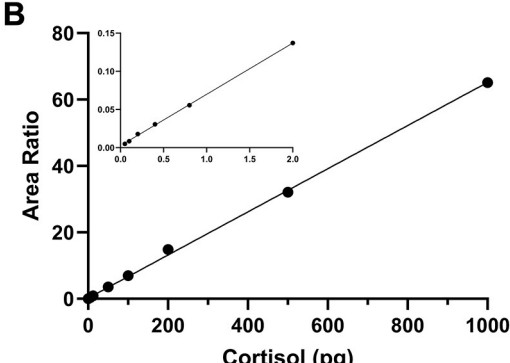

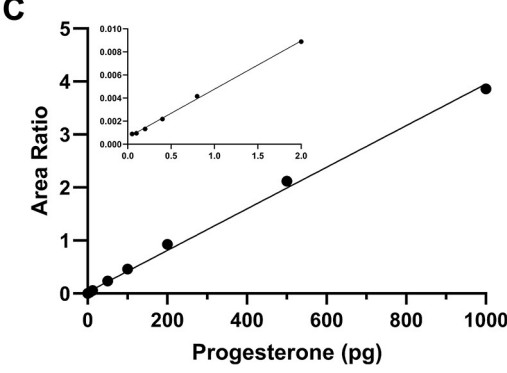

**Fig 2.** Calibration curves for the 3 most abundant steroids in human primary teeth **(A)** DHEA, **(B)** cortisol, and **(C)** progesterone. Area ratio is calculated by dividing the peak area of the analyte by the peak area of the corresponding deuterated internal standard in the same sample. Calibration curve range was 2 to 1000 pg per tube (0.91 to 18,182 pg/ml) for DHEA and 0.05 to 1000 pg per tube (36.4 to 18,182 pg/ml) for cortisol and progesterone. Insets display the lowest standards on the calibration curves and demonstrate the excellent assay sensitivity. The line of fit equations are $y = 0.0477x + 0.0802$, $y = 0.0650x + 0.1729$, $y = 0.0039x + 0.0250$ for DHEA, cortisol, and progesterone, respectively.

expected, there was a significant positive correlation between androstenedione and testosterone in males and females, as well as both sexes combined (Fig 5). Also as expected, there were significant positive correlations between glucocorticoids (cortisol, cortisone, corticosterone, DHC) (Fig 6). Cortisol and cortisone were significantly positively correlated in the male samples. Cortisol and corticosterone, cortisol and DHC, cortisone and DHC, and corticosterone and DHC were significantly positively correlated in the female samples. All

**Table 4. Absence of matrix effects.** Parallelism was assessed by comparing the slopes of the serially diluted teeth and the calibration curves (in neat solution).

| Steroid | Slope (Neat) | Slope (Teeth) | %Δ |
|---|---|---|---|
| DHEA | 0.307 | 0.308 | 0.3 |
| Cortisol | 0.044 | 0.043 | -1.4 |
| Progesterone | 0.047 | 0.048 | 0.6 |
| Testosterone | 0.076 | 0.089 | 16.2 |

glucocorticoids (GCs) were significantly positively correlated with each other when both sexes were combined.

## Discussion

Here, we developed and validated a novel method to measure multiple steroids in human primary teeth via LC-MS/MS. The assay shows excellent specificity, sensitivity, accuracy, and precision, with minimal or no matrix effects. We then used this assay to perform steroid profiling in human primary teeth and were able to quantify 8 steroids: DHEA, cortisol, progesterone, cortisone, corticosterone, DHC, androstenedione, and testosterone. Steroid levels were generally low in primary teeth, but DHEA, cortisol, and progesterone levels were higher than those of other steroids examined. Finally, we compared steroid concentrations between male and female primary bottom central incisors and found no significant sex differences. As expected, we found significant positive correlations between androgens (androstenedione and testosterone) and between GCs (cortisol, cortisone, corticosterone, DHC). This is the first study to measure steroids in human primary teeth using LC-MS/MS. This method could be useful for the assessment of ELS and endocrine disorders or disruption during early development, including the prenatal period.

We validated our LC-MS/MS method through the assessment of specificity, sensitivity, accuracy, precision, and matrix effects. Our method ensures specificity by using scheduled multiple reaction monitoring with two unique mass transitions and unique in retention times for each steroid. The signal to noise ratios for blanks and double blanks were below 3, confirming the absence of assay interference from assay solvents and reagents. The high specificity of the LC-MS/MS circumvents a common problem with immunoassays: antibody cross-reactivity, especially at low analyte levels [52,53]. Sensitivity is also several-fold higher than many immunoassays [54,55]. The calibration curves span a wide range from as low as 0.05 pg to 1000 pg per tube, and they remain linear at the low end of the range. This is critical because the steroid amounts in an entire primary tooth were generally quite low. All steroids showed

**Table 5. Recovery of exogenous steroids from human primary teeth.** Recovery was assessed by subtracting unspiked tissue pool values from spiked tissue pool values and dividing by the amount of steroid added.

| Steroid | % Recovery | % CV |
|---|---|---|
| DHEA | 101.1 | 5.6 |
| Cortisol | 101.7 | 13.3 |
| Progesterone | 98.9 | 5.5 |
| Cortisone | 95.1 | 6.3 |
| Corticosterone | 103.3 | 5.2 |
| Androstenedione | 105.3 | 6.2 |
| DHC | 100.6 | 9.1 |
| Testosterone | 99.6 | 6.8 |

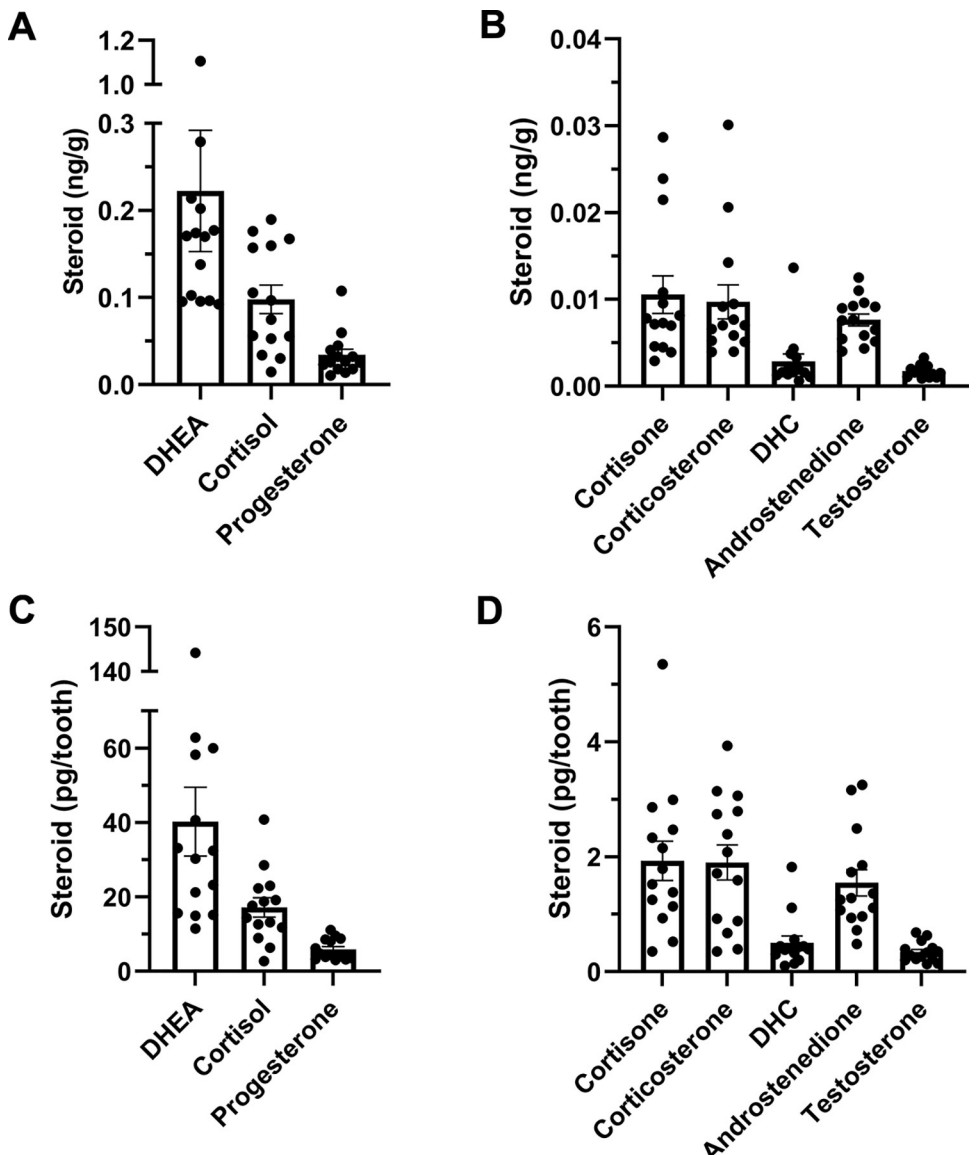

**Fig 3.** Concentrations of **(A)** the 3 most abundant steroids (in ng/g) and **(B)** the 5 other steroids that were quantifiable (in ng/g) in human primary teeth. Absolute steroid levels (pg) per tooth of **(C)** the 3 most abundant steroids and **(D)** the other 5 steroids quantified in human primary teeth. Steroid levels in primary teeth were generally low (e.g. ~15 pg cortisol/tooth and ~0.5 pg of testosterone/tooth). Data are shown as mean ± *SEM*. $n$ = 14 teeth.

high accuracy and precision, as assessed via quality controls. The lack of matrix effects is demonstrated by the internal standards, parallelism of serial-diluted samples to the calibration curves, and excellent recovery of exogenous steroids.

Cortisol is the only steroid that has been previously quantified in human teeth [40–42]. Nejad et al. (2016) measured cortisol in wisdom teeth using a commercial salivary cortisol ELISA and reported high levels (~8000 ng/g), compared to the levels seen here (~0.1 ng/g). While Nejad et al. (2016) reported cleaning the tooth outer surfaces, they did not report cleaning the pulp chambers, and thus their levels might reflect cortisol from residual blood and/or pulp. Here, we thoroughly cleaned tooth outer surfaces and pulp chambers. Furthermore, the ELISA used by Nejad et al (2016) was designed for human saliva and was not validated for

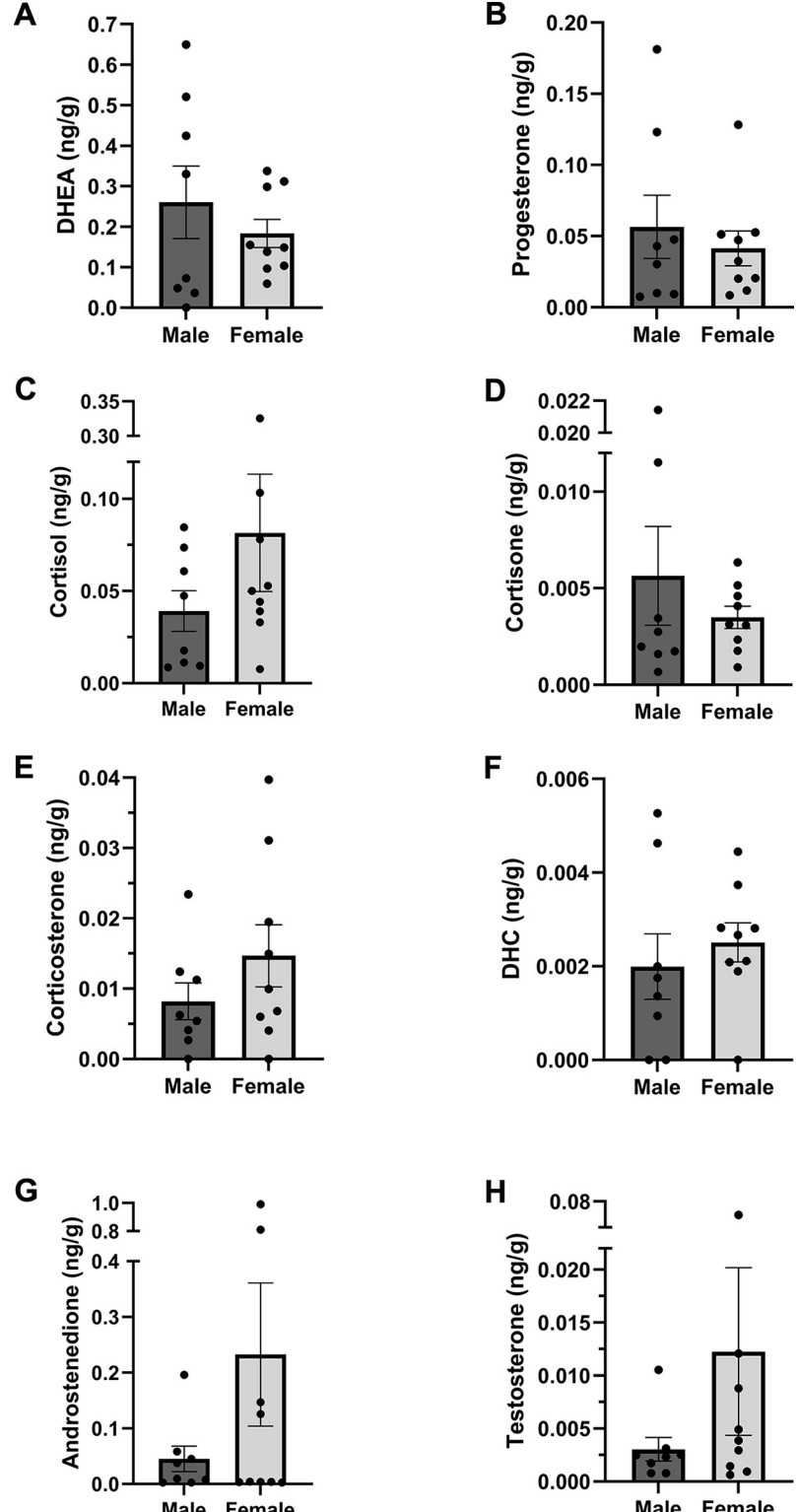

**Fig 4.** Human primary teeth **(A)** DHEA, **(C)** progesterone, **(C)** cortisol, **(D)** cortisone, **(E)** corticosterone, **(F)** 11-dehydrocorticosterone (DHC), **(G)** androstenedione, and **(H)** testosterone levels from male and female subjects (approximately 5–6 yr of age). No significant sex differences were observed in any of the quantifiable steroids. Data are shown as mean ± *SEM*. *n* = 8 males and 9 females.

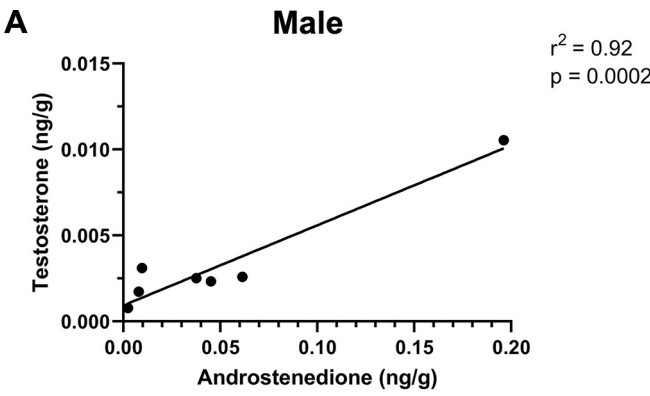

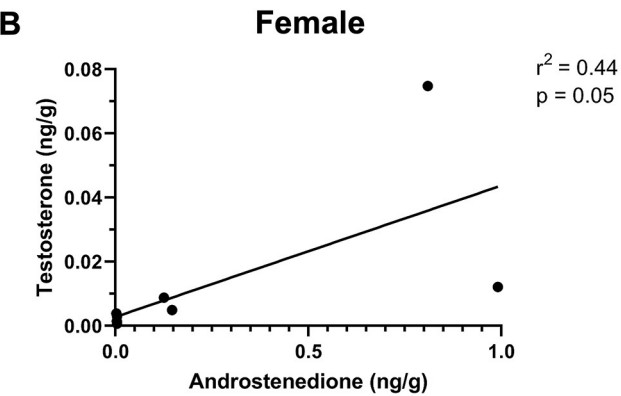

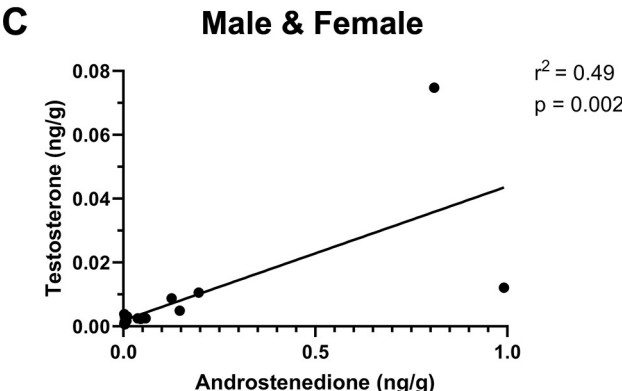

**Fig 5.** Simple linear regression of testosterone vs. androstenedione in primary teeth of (**A**) male, (**B**) female, and (**C**) combined sexes. Levels of testosterone and androstenedione, which are both androgens, were significantly positively correlated. Data were analyzed using Spearman's rho. $n$ = 8 males and 9 females.

teeth. Thus, it is unclear how the matrix may have affected the observed values. Moreover, immunoassays are subject to antibody cross-reactivity [52–55]. Quade et al. (2021, 2023) used the same ELISA to measure cortisol in archeological adult teeth and archeological and modern primary teeth [40,42]. Their reported cortisol concentrations in modern primary teeth ranged from 0.07 to 1.8 ng/g, more similar to our results. However, only 25% of modern primary

**A**  **Male**

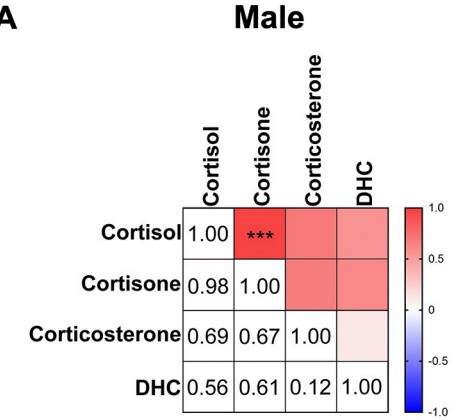

**B**  **Female**

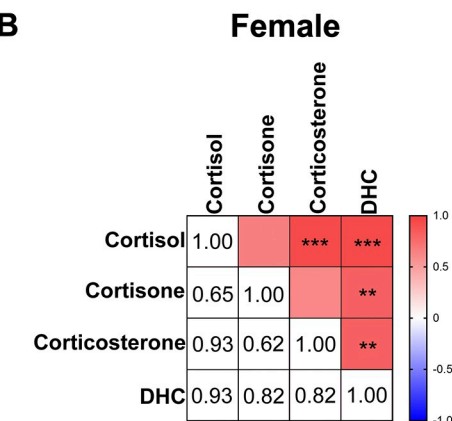

**C**  **Male & Female**

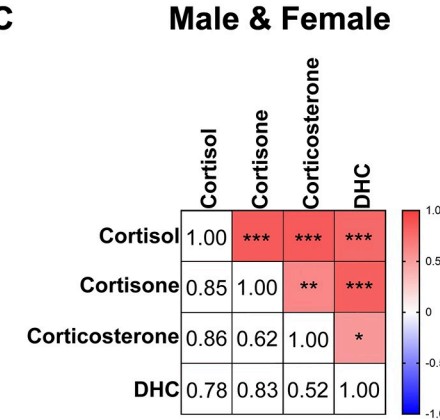

**Fig 6.** Matrices of pairwise Spearman rho correlations between all glucocorticoid levels in primary teeth for **(A)** male, **(B)** female, and **(C)** combined sexes. Positive correlations are denoted in red, and negative correlations are denoted in blue. The r values are shown in the bottom left section, and the statistical significance levels are shown in the top right section. Levels of glucocorticoids were significantly positively correlated, particularly when the sexes were combined and the sample size was higher. $n$ = 8 males and 9 females. *$p \leq 0.05$; **$p \leq 0.01$; ***$p \leq 0.001$.

teeth had detectable cortisol in Quade et al. (2023), likely because the ELISA has a lower limit of quantification of 3 pg per sample. Our LC-MS/MS assay can quantify as little as 0.05 pg per sample, and cortisol was detectable in 100% of primary teeth.

We were able to quantify 8 steroids in human primary teeth: DHEA, cortisol, progesterone, cortisone, corticosterone, DHC, androstenedione, and testosterone. DHEA, cortisol, progesterone were the most abundant steroids in primary teeth, and these 3 steroids are all secreted by the adrenal glands. Cortisol and progesterone play critical roles in gestation and fetal development and circulate at high levels during pregnancy [56–60]. High levels of cortisol are secreted by the fetal adrenals in late pregnancy to stimulate surfactant synthesis in the lungs [57,61,62]. Large amounts of progesterone are secreted throughout gestation by the corpus luteum and then by the placenta, to maintain the uterine lining [60,63,64]. Of the steroids examined, DHEA was the most abundant in primary teeth. DHEA and DHEA-sulfate are secreted by the zona reticularis of the adrenals and are also the most abundant steroids in the blood, saliva, and hair [45,65–67].

Overall, steroid concentrations in primary teeth are quite low. There are several possible explanations. First, only free steroids in the blood might be incorporated into primary teeth, and most circulating steroids are bound to binding globulins, such as CBG [68–70]. Second, teeth undergo little remodeling once mineralization is complete [28–30,37]. Thus, there might be a limited window for steroid deposition. Third, steroid concentrations may differ with tooth type. Primary bottom central incisors, the tooth type used in our study of sex differences, are the first teeth to complete formation [28–31]. Canines and molars, which complete formation later, may have a longer window for steroid deposition [28–31]. However, various drugs have been detected in adult teeth in post-mortem analysis, suggesting that deposition of compounds is possible after tooth formation is complete [71,72]. Fourth, steroid concentrations might differ between tooth compartments. Here, whole teeth were analyzed, but the concentrations of some compounds, including cortisol, are higher in dentin than in enamel, as dentin is less mineralized and allows for easier deposition of organic molecules [42,71]. Lastly, the low steroid concentrations are not likely due to steroid breakdown, as steroids are stable for long periods in teeth, feathers, hair, nails, and baleen [43–49].

No significant sex differences in steroids were detected in the primary bottom central incisors. If steroids are deposited during tooth formation (from week 6 of gestation to 18–36 months after birth) then primary teeth reflect steroid levels before puberty, which may account for the lack of sex differences [28–31]. Surprisingly, testosterone showed a suggestion to be higher in females than in males, but this was not statistically significant. More samples should be examined to see if this pattern is replicable.

Several future investigations would be useful. It is possible that steroids in teeth exist as conjugated metabolites, as in urine [73–75]. Thus, conjugated steroids in primary teeth can be examined. Second, measurement of sex steroids in adult teeth and possible sex differences would be informative. Third, it would be of interest to measure GC levels in primary teeth and associations with developmental stressors such as maternal illness and low socioeconomic status [22,76–78]. Fourth, tooth GC levels can be correlated with growth marks in primary teeth, which have also been used to assess ELS [32,33]. Additionally, teeth can be separated at these growth marks to allow for temporal resolution [27,28]. For example, by separating teeth at the neonatal line, the growth mark that denotes an individual's birth, we can tease apart prenatal and postnatal GC levels [25,26]. Such studies would shed light on the potential of using primary teeth GCs to assess ELS and other developmental conditions.

Here, we developed a method to simultaneously measure multiple steroids in human primary teeth. We examined 17 steroids and could quantify 8 steroids in primary teeth. Our LC-MS/MS method provides high specificity and sensitivity, which are critical because dental steroid levels are low. Primary teeth might be an integrated long-term measure of steroids in the blood and thus provide a way to assess ELS and provide a window into the prenatal environment.

## Acknowledgments

We thank Dr. Chunqi Ma for assistance with experimental design, data collection and data analysis; Dr. Benjamin Pliska for the classification of tooth samples; and the UBC Genotyping Facility.

## Disclosure summary

The authors have nothing to disclose.

## Author Contributions

**Conceptualization:** Ruolan S. Wu, Jordan E. Hamden, Melody Salehzadeh, Kim L. Schmidt, Michael S. Kobor, Kiran K. Soma.

**Data curation:** Ruolan S. Wu.

**Formal analysis:** Ruolan S. Wu.

**Funding acquisition:** Michael S. Kobor, Kiran K. Soma.

**Investigation:** Ruolan S. Wu, Jordan E. Hamden, Melody Salehzadeh, Michael X. Li.

**Methodology:** Asmita Poudel.

**Resources:** Kiran K. Soma.

**Supervision:** Kiran K. Soma.

**Writing – original draft:** Ruolan S. Wu.

**Writing – review & editing:** Ruolan S. Wu, Jordan E. Hamden, Melody Salehzadeh, Michael X. Li, Asmita Poudel, Kim L. Schmidt, Michael S. Kobor, Kiran K. Soma.

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
