## [Decision Letter · Decision Letter 0]

14 May 2024

PONE-D-24-09259Steroid profiling in human primary teeth via liquid chromatography-tandem mass spectrometry for long-term retrospective steroid measurementPLOS ONE

Dear Dr. Wu, Thank you for submitting your manuscript to PLOS ONE. After careful consideration, we feel that it has merit but does not fully meet PLOS ONE’s publication criteria as it currently stands. Therefore, we invite you to submit a revised version of the manuscript that addresses the points raised during the review process. Specifically, please focus on the methodologies related to extraction and LC-MS/MS. It is important that you include the details of LC-MS/MS method validation. The differential levels of testosterone and their clinical significance should be explained.

We look forward to receiving your revised manuscript.

Kind regards,

Subrata Deb

Academic Editor

PLOS ONE

Journal Requirements:

"Natural Sciences and Engineering Research Council of Canada (NSERC) Discovery Grant (RGPIN-2019-04837) and Discovery Accelerator Supplement (RGDAS-2019-00033) to KKS; UBC Grants for Catalysing Research Clusters (F18-05858), and a New Frontiers in Research Fund Grant (F21-04739) to MSK; Michael Smith Health Research BC and CLEAR Foundation Postdoctoral Fellowship to JEH; and NSERC CGS-M and CGS-D fellowships to MS."

Reviewers' comments:

Reviewer's Responses to Questions

**Comments to the Author**

1. Is the manuscript technically sound, and do the data support the conclusions?

Reviewer #1: Partly

Reviewer #2: Yes

2. Has the statistical analysis been performed appropriately and rigorously? 

Reviewer #1: Yes

Reviewer #2: Yes

3. Have the authors made all data underlying the findings in their manuscript fully available?

Reviewer #1: Yes

Reviewer #2: No

4. Is the manuscript presented in an intelligible fashion and written in standard English?

Reviewer #1: Yes

Reviewer #2: Yes

5. Review Comments to the Author

Reviewer #1: The manuscript is well written and the method is extensively described. I have, however, a few unclear issues.

1. Was the use of polypropylene tubes unavoidable in the extraction procedure? I am asking this since plastic contaminants can increase background in MS/MS. Did blank samples go through the same extraction procedures as the tooth powder?

2. At what point were the internal standards added to the sample or blank? Did the internal standards also go through the extraction procedure?

3. What were the LOQ for the analytes of interest?

4. What was the volume for the calibration curves? You mention that the standard curve started at 0.05 pg, this seems extremely low, what was the minimal concentration used?

5. What was the number of replicates for intra-assay CV analysis? What about the inter-assay CV, how many independent analyses were included in the calculation? Same question for recovery analysis and linearity.

6. Why are there higher levels of Testosterone for female samples? Testosterone is still lower even in infant girls than boys. Can the higher levels of Testo in female primary teeth be explained? You mention that male fetuses are exposed to surging levels of Testo for several weeks, how are they still lower than female levels?

7. Minor comment: on page 14 line 247 I think is meant to cite Table 3.

Reviewer #2: General Comments

I find the manuscript to be of interest to other researchers in the field. The paper presents a new method to quantify steroid level in human primary teeth. The study is interesting and easy to read. The authors must please report the method validation according to analytical chemistry specifications (see specific comments). I am also concerned that the reported values where calculated correctly (see specific comments)

If the authors can please just check and correct the calculations, the manuscript can be accepted for publication.

Headings should be numbered.

Specific feedback to the authors is given here, if they wish to improve the manuscript:

Introduction:

Line 67

…primary teeth’s chemical composition.

Development and Validation

Line 91 – please add the age range.

Table 1 – what significance level are you using for your p value?

Steroid Analysis by LC-MS/MS

Line 143 - 45 ul is a quite a large injection volume for an UPLC, especially considering you reconstituted in 55 ul and then also only transferred the supernatant to the LC vials. I am not sure if you were able to inject all 45 ul. Did you check and adjust the autosampler needle?

Line 153 – the needle wash on these intruments are usually a strong and weak wash. Just check the method.

Line 164 – please give calibration range, ie what is the volulme of the tube (pg per tube is not meaningful to readers).

Line 195 – what is meant by “vehicle”?

Results

Please check figure 2 – calibration curves should not go through 0 (I can’t clearly tell from the image). Please add line of fit equation on the figures.

The reporting of the calibration curve ranges are confusing.

Lines 226 – 229 – the range is 0.05 to 1000 pg or 2 to 1000 pg depending on the steroids.

However, in Fig 2’s legend you are reporting 2 to 1000 or 0.05 to 1000 pg per tube.

The authors mention gram amounts, I am concerned that the gram amount was directly read off the calibration curve for the samples. This is not correct as the final sample volume and the standards’ tube volume (from what I can tell) is not the same.

Please confirm that the correct calculations were done.

Method validation (for analytical methods) should be reported as %RSD, recovery, LODs and LOQs for all analytes, please add.

SEX DIFFERENCES

Line 297 – please define “GCs”

Discussion

Please add is the levels detected of concern. Why is some levels higher for some teeth?

6. PLOS authors have the option to publish the peer review history of their article (what does this mean?). If published, this will include your full peer review and any attached files.

Reviewer #1: No

Reviewer #2: **Yes: **Madelien Wooding

---

## [Author Response · Author response to Decision Letter 0]

18 Jun 2024

Manuscript Number: PONE-D-24-09259

We thank the reviewers for their helpful comments and suggestions that allowed us to greatly improve this manuscript. We have addressed every comment point by point, as detailed below and in our rebuttal letter attached at the end. Changes in the revised manuscript are in blue (in the rebuttal letter). Our revisions focused on clarifying methodologies for steroid extraction and steroid quantification by LC-MS/MS. In particular, we have revised Table 3 to include LLOQs and LODs and revised Figure 2. Additional details about our methods and participants have been provided in the Methods section. We also deposited our laboratory protocols in protocols.io (DOI: dx.doi.org/10.17504/protocols.io.e6nvw1b3dlmk/v1). Lastly, we revised the section of the Discussion about testosterone levels.

Reviewer #1:

Reviewer #1: The manuscript is well written and the method is extensively described. I have, however, a few unclear issues.

1. Was the use of polypropylene tubes unavoidable in the extraction procedure? I am asking this since plastic contaminants can increase background in MS/MS. Did blank samples go through the same extraction procedures as the tooth powder?

Response: We avoided using plastics, such as polypropylene, in the extraction procedure as much as possible. However, tubes for the homogenization are made from polypropylene. Also, for the final step of the extraction process, we centrifuge at a high speed (over 16,000× g) and this requires polypropylene microcentrifuge tubes. However, the contact time of samples with these microcentrifuge tubes is minimized (~7 min total). We agree with the reviewer that plastic contaminants can increase background; however, our method showed high selectivity for the target analytes, as double blank samples did not contain any interferences in the monitored analytes and internal standard MRM transitions.

Yes, blanks and double blanks went through the same extraction procedure as the tooth samples, and no signals were observed (Fig. 1).

2. At what point were the internal standards added to the sample or blank? Did the internal standards also go through the extraction procedure?

Response: The internal standards were added to the calibration curves, quality controls, blanks, and samples (but not the double blanks) prior to homogenization and steroid extraction. Therefore, the internal standards went through the entire extraction procedure. We have now clarified this point in the Methods section (line 176 and lines 179-181).

“Prior to homogenization, 50 μl of deuterated internal standards (dehydroepiandrosterone-d6, testosterone-d5, cortisol-d4, corticosterone-d8, progesterone-d9, pregnenolone-d4, allopregnanolone-d4, 17β-estradiol-d4, and aldosterone-d7 C/D/N Isotopes Inc., Pointe-Claire, Canada) in 50:50 HPLC-grade methanol:MilliQ water were added to the calibration curves, quality controls, blanks, and samples (40 pg/sample of DHEA-d6, pregnenolone-d4, allopregnanolone-d4, and 20 pg/sample of all other internals standards).”

3. What were the LOQ for the analytes of interest?

Response: The LLOQs were the lowest concentration on the calibration curve (2.0 pg/10 μl for DHEA, pregnenolone, and allopregnanolone, and 0.05 pg/10 μl for all other steroids) within 20% of the nominal concentration.

We have now revised Table 3 to include LLOQs.

4. What was the volume for the calibration curves? You mention that the standard curve started at 0.05 pg, this seems extremely low, what was the minimal concentration used?

Response: For each point on the calibration curves, we used 10 μl of steroid in 50:50 HPLC-grade methanol:MilliQ water. The concentrations of the standards ranged from 0.005 pg/μl to 100 pg/μl. Therefore, the calibration curve ranged from 0.05 to 1000 pg per tube. We agree that the LLOQs are extremely low, making the method very sensitive.

5. What was the number of replicates for intra-assay CV analysis? What about the inter-assay CV, how many independent analyses were included in the calculation? Same question for recovery analysis and linearity.

Response: Intra- and inter-assay quality controls were assessed in triplicate for each run. Inter-assay variation (%CV) was calculated from the results from 4 independent assays. To assess recovery, extracts from 6 teeth were pooled and split into 2 groups, either spiked with known amounts of steroid or unspiked, with 5 replicates per group. Linearity was assessed in singleton.

6. Why are there higher levels of Testosterone for female samples? Testosterone is still lower even in infant girls than boys. Can the higher levels of Testo in female primary teeth be explained? You mention that male fetuses are exposed to surging levels of Testo for several weeks, how are they still lower than female levels?

Response: Statistical analyses showed no significant difference in testosterone between males and females (p = 0.29). The non-significant trend for higher testosterone levels in female samples than in male samples was unexpected to us. However, at this time, we conclude that there was no significant difference in testosterone levels in primary teeth from males and females. Future studies may examine more samples to see if this pattern is replicable.

We have revised the Discussion (lines 395-397).

“No significant sex differences in steroids were detected in the primary bottom central incisors. If steroids are deposited during tooth formation (from week 6 of gestation to 18-36 months after birth) then primary teeth reflect steroid levels before puberty, which may account for the lack of sex differences [28-31]. Surprisingly, testosterone showed a suggestion to be higher in females than in males, but this was not statistically significant. More samples should be examined to see if this pattern is replicable.”

7. Minor comment: on page 14 line 247 I think is meant to cite Table 3.

Response: This has been corrected in the Results section (line 254-255).

“Intra-assay and inter-assay variation (% CV) were below 5%, confirming the precision of the assay (Table 3).”

Reviewer #2

Reviewer #2: General Comments

I find the manuscript to be of interest to other researchers in the field. The paper presents a new method to quantify steroid level in human primary teeth. The study is interesting and easy to read. The authors must please report the method validation according to analytical chemistry specifications (see specific comments). I am also concerned that the reported values where calculated correctly (see specific comments)

If the authors can please just check and correct the calculations, the manuscript can be accepted for publication.

Headings should be numbered.

Response: The headings were formatted to follow PLOS ONE's style requirements, as per the instructions to authors on the journal website. Thus, instead of numbering, we have formatted them into level 1, 2, and 3 headings, in bold type 18pt, 16pt, and 14pt font, respectively.

Introduction:

Line 67

…primary teeth’s chemical composition.

Response: We have corrected this in the Introduction section (line 67).

“Environmental exposures are also reflected in primary teeth’s chemical composition [27, 28, 35, 36].”

Development and Validation

Line 91 – please add the age range.

Response: We added this information to the Development and Validation section (line 92-93).

“Human primary teeth were from male and female participants ranging from 4 to 8 years of age.”

Table 1 – what significance level are you using for your p value?

Response: Significance criterion was set at p ≤ 0.05. We have now clarified this as a note in Table 1.

“Note: Data are shown as mean ± SEM. n = 8 males and 9 females. Significance criterion was set at p ≤ 0.05.”

Steroid Analysis by LC-MS/MS

Line 143 - 45 ul is a quite a large injection volume for an UPLC, especially considering you reconstituted in 55 ul and then also only transferred the supernatant to the LC vials. I am not sure if you were able to inject all 45 ul. Did you check and adjust the autosampler needle?

Response: The volume of the pellet was negligible; therefore, nearly all of the 55 μl was transferred and available for injection (50-55 μl). The volume remaining after injection was assessed and shown to be 5-10 μl.

We are completely confident that we were able to inject all 45 μl here and in our previous papers [1,2].

Line 153 – the needle wash on these instruments are usually a strong and weak wash. Just check the method.

Response: The instrument setting was optimized for needle wash using 100% methanol. This wash was effective in preventing carryover (i.e. cross-contamination between samples). Neat solvents run after needle wash did not contain MRM transitions matching with the analytes and internal standards.

Line 164 – please give calibration range, ie what is the volulme of the tube (pg per tube is not meaningful to readers).

Response: The volume of the calibration curves was 10 μl. The concentration of the standards ranged from 0.005 pg/μl to 100 pg/μl. Therefore, the calibration curve ranged from 0.05 to 1000 pg per tube.

Line 195 – what is meant by “vehicle”?

Response: We have removed the term vehicle from the manuscript (line 198).

“The tubes were split into two groups (n = 6/group) and spiked with either known amounts of steroid (based on expected steroid concentrations; 50 pg of DHEA, 5 pg of cortisol, and 2 pg of all other steroids) or unspiked.”

Results

Please check figure 2 – calibration curves should not go through 0 (I can’t clearly tell from the image). Please add line of fit equation on the figures.

The reporting of the calibration curve ranges are confusing.

Response: We have now revised Figure 2. The point at the origin has been removed, and the line of fit equations of the calibration curves have been added to the legend.

“Fig 2. Calibration curves for the 3 most abundant steroids in human primary teeth (A) DHEA, (B) cortisol, and (C) progesterone. Area ratio is calculated by dividing the peak area of the analyte by the peak area of the corresponding deuterated internal standard in the same sample. Calibration curve range was 2 to 1,000 pg per tube for DHEA and 0.05 to 1,000 pg per tube for cortisol and progesterone. Insets display the lowest standards on the calibration curves and demonstrate the excellent assay sensitivity. The line of fit equations are y = 0.0477x + 0.0802, y = 0.0650x + 0.1729, y = 0.0039x + 0.0250 for DHEA, cortisol, and progesterone, respectively.”

Lines 226 – 229 – the range is 0.05 to 1000 pg or 2 to 1000 pg depending on the steroids. However, in Fig 2’s legend you are reporting 2 to 1000 or 0.05 to 1000 pg per tube.

Response: Steroids such as DHEA, pregnenolone, and allopregnanolone, show lower ionization efficiencies, and thus, assay sensitivity is lower. Instead of a lower limit of quantification at 0.05 pg, their lower limit of quantification is at 2 pg. Therefore, in Figure 2, DHEA is reported to range from 2 to 1000 pg per tube, while cortisol and progesterone range from 0.05 to 1000 pg per tube.

The authors mention gram amounts, I am concerned that the gram amount was directly read off the calibration curve for the samples. This is not correct as the final sample volume and the standards’ tube volume (from what I can tell) is not the same. Please confirm that the correct calculations were done.

Response: In this manuscript, we reported the level of steroids in teeth as ng/g or pg/tooth. The raw data were obtained in pg, using a calibration curve that ranged from 0.05 to 1000 pg per tube. The concentrations of the standards ranged from 0.005 pg/μl to 100 pg/μl. For each point on the calibration curves, we used 10 μl of steroid in 50:50 HPLC-grade methanol:MilliQ water. For each tube, steroids from an entire tooth were assessed. The obtained quantitative value in pg was then converted to ng/g, based on the weight of each tooth. For steroid analysis in teeth, final results in ng/g are more suitable and useful as samples are in solid form. In our previous work, as well as other various published work on tissues, data are reported in ng/g [1,2].

Method validation (for analytical methods) should be reported as %RSD, recovery, LODs and LOQs for all analytes, please add.

Response: We have revised Table 3 to include the LLOQs and LODs in the note below the table. Inter- and intra- assay %CVs (%RSD) were already reported in Table 3, and recovery has already been reported in Table 5.

SEX DIFFERENCES

Line 297 – please define “GCs”

Response: This has now been defined in the Results section (line 305).

“All glucocorticoids (GCs) were significantly positively correlated with each other when both sexes were combined.”

Discussion

Please add is the levels detected of concern. Why is some levels higher for some teeth?

Response: This comment is unclear to us, but we believe it may be asking about the clinical significance of the dental steroid levels, and whether these differences can be explained.

Our study is the first to examine multiple steroids in human primary teeth by LC-MS/MS, so the clinical significance awaits further study. Since various organic and inorganic compounds have been detected in primary teeth, it is hypothesized that circulating steroids may be deposited during tooth formation, which begins in utero. Thus, elevated steroids may reflect pathologies involving the dysregulation of steroid hormone secretion during perinatal life.

We discuss this point in paragraphs 3-4 of the Introduction (lines 60-77). We also suggest studies in the Discussion to further investigate this hypothesis. Specifically, correlation studies between steroid levels in primary teeth and developmental stressors such as maternal illness or low socioeconomic status (lines 401-403). 

References

1. Hamden JE, Gray KM, Salehzadeh M, Kachkovski GV, Forys BJ, Ma C, Austin SH, Soma KK. Steroid profiling of glucocorticoids in microdissected mouse brain across development. Dev Neurobiol. 2021;81(2):189-206. Epub 2021/01/10. doi: 10.1002/dneu.22808. PubMed PMID: 33420760.

2. Jalabert C, Ma C, Soma KK. Profiling of systemic and brain steroids in male songbirds: Seasonal changes in neurosteroids. J Neuroendocrinol. 2021 Jan;33(1):e12922. doi: 10.1111/jne.12922. Epub 2020 Dec 14. PMID: 33314446.

---

## [Decision Letter · Decision Letter 1]

5 Jul 2024

PONE-D-24-09259R1Steroid profiling in human primary teeth via liquid chromatography-tandem mass spectrometry for long-term retrospective steroid measurementPLOS ONE

Dear Dr. Wu,

Thank you for submitting your manuscript to PLOS ONE. After careful consideration, we feel that it has merit but does not fully meet PLOS ONE’s publication criteria as it currently stands. Therefore, we invite you to submit a revised version of the manuscript that addresses the points raised during the review process.

We look forward to receiving your revised manuscript.

Kind regards,

Tommaso Lomonaco, Ph.D

Academic Editor

PLOS ONE

Additional Editor Comments:

Dear Authors, according to the reviewers comments I suggest a major revisions. Please take care of the comments from the 2 reviewer.

Best regards,

Tommaso Lomonaco

Reviewers' comments:

Reviewer's Responses to Questions

**Comments to the Author**

1. If the authors have adequately addressed your comments raised in a previous round of review and you feel that this manuscript is now acceptable for publication, you may indicate that here to bypass the “Comments to the Author” section, enter your conflict of interest statement in the “Confidential to Editor” section, and submit your "Accept" recommendation.

Reviewer #1: All comments have been addressed

Reviewer #2: (No Response)

2. Is the manuscript technically sound, and do the data support the conclusions?

Reviewer #1: Yes

Reviewer #2: Partly

3. Has the statistical analysis been performed appropriately and rigorously? 

Reviewer #1: Yes

Reviewer #2: Yes

4. Have the authors made all data underlying the findings in their manuscript fully available?

Reviewer #1: Yes

Reviewer #2: Yes

5. Is the manuscript presented in an intelligible fashion and written in standard English?

Reviewer #1: Yes

Reviewer #2: Yes

6. Review Comments to the Author

Reviewer #1: My concerns have been answered in a suitable manner. The method is more clear and reliable to me now.

Reviewer #2: The authors have still not provided the calculations to determine the concentrations of the targets in the tooth samples. There is not enough information given in the paper for me to check if the calculations are correct. With the information given I am struggling to agree with the concentration values reported.

LODs and LOQs need to be calculated with standard methods and how they were calculated need to be added in the manuscript. Also, these will referer to the insturments capabilities and not the method as calibration curves were done on pure standards; method matched/ standard addition methods were not done.

7. PLOS authors have the option to publish the peer review history of their article (what does this mean?). If published, this will include your full peer review and any attached files.

Reviewer #1: No

Reviewer #2: **Yes: **Madelien Wooding

---

## [Author Response · Author response to Decision Letter 1]

12 Jul 2024

Manuscript Number: PONE-D-24-09259R1

We thank Reviewer #2 for their helpful comments and suggestions. We have addressed every comment point by point as detailed below, and as detailed in the attached document titled "Response to Reviewers" in which changes in the revised manuscript are in blue. Our revisions focused on clarifying the calibration curve and the calculation of sample concentrations. In particular, Table 3 has been revised further to clarify how we determined the LLOQs and LODs. Additional details have been provided in the Methods and Results sections.

Reviewer #2

Steroid Analysis by LC-MS/MS

(please note for future analysis it is recommended to use MS grade solvents and not HPLC grade

for mass spec)

Response: Thank you for this suggestion. We have compared MS-grade and HPLC-grade solvents in the past but will revisit this point in the future, if necessary.

Line 169 – Please add the amount of liquid added to the standard. – the 10ul.

Response: 

The initial amount of liquid added to the standards (10 μl) has been added to the Methods section (line 166-174). Note that these standards in the calibration curve are processed in the same way as the unknowns (i.e. teeth samples) and resuspended in 55 μl prior to injection. This important point is now clearer in the Methods.

“Standards ranged from 0.005 to 100 pg/μl. 10 μl of each standard was used to make the calibration curves. The calibration curves were then processed alongside the samples under the same extraction procedure. Both the calibration curves and the teeth samples were dried in a vacuum centrifuge and resuspended in the same final volume of 55 μl of 1:3 HPLC-grade methanol:MilliQ water. The calibration curve range was 0.05 to 1000 pg per tube (0.91 to 18,182 pg/ml) for all steroids (excluding DHEA, pregnenolone, and allopregnanolone) and prepared in 50:50 HPLC-grade methanol:MilliQ water. The calibration curve range was 2 to 1000 pg per tube (36.4 to 18,182 pg/ml) for DHEA, pregnenolone, and allopregnanolone.”

Also, note you have now reconstituted your teeth samples in 55ul and the standards in 10ul, thus it is not directly comparable.

For example, if for a sample you read off the calibration curve you detected 2ng that is 2ng/10ul = 0.2ng/ul.

Your sample was reconstituted into 55ul

So, 0.2ng/ul = 11ng/55ul

Thus, you had 11 ng in the original sample.

This you can then report per the gram of sample weighed.

Response: We have revised the Methods to make it clearer that teeth samples and standards were all reconstituted in 55 μl (line166-174). We have used similar protocols in many previous papers, including our most recent paper in Nature [1]. We apologize if this detail was unclear in the previous version of the manuscript.

Also, what was the injection volume for the standards as it can not be 45ul seeing you made them in 10ul?

Response: Both the standards and the teeth samples were reconstituted in the same volume of 55 μl. 10 μl is the initial volume of standard added to the tube. Standards were then processed alongside the teeth samples using the same extraction process. Therefore, both the standards and the teeth samples were treated the same way – dried in a vacuum centrifuge and resuspended in 55 μl of 1:3 HPLC-grade methanol:MilliQ water. The injection volume was 45 μl for both the standards and the teeth samples.

We have used this method in numerous published studies [1-5]. In previous studies and the present study, the quality controls and recovery values were as expected. We hope these points reassure the reviewer with regard to our calculations.

Line 176 – Add prior to homogenization and extraction the IS was added…

Response: We have revised this in the Methods section (line 182).

“Prior to homogenization and extraction, 50 μl of deuterated internal standards (dehydroepiandrosterone-d6, testosterone-d5, cortisol-d4, corticosterone-d8, progesterone-d9, pregnenolone-d4, allopregnanolone-d4, 17β-estradiol-d4, and aldosterone-d7 C/D/N Isotopes Inc., Pointe-Claire, Canada) in 50:50 HPLC-grade methanol:MilliQ water were added to the calibration curves, quality controls, blanks, and samples (40 pg/sample of DHEA-d6, pregnenolone-d4, allopregnanolone-d4, and 20 pg/sample of all other internals standards).”

Results

Lines 232 – the calibration range is not 2 to 1000 pg as you could not possible inject all the standard (ie the entire for example 2ng on the column). Please report the calibration range as a

concentration. Also add the injection amount to give the gram on column per injection.

Fig 2’s legend you are reporting 2 to 1000 or 0.05 to 1000 pg per tube please report per 10ul.

Response: We have revised the Results section to include the concentration of the calibration curve range and the injection amount (line 235-242).

“The calibration curve consisted of 13 points ranging from 0.05 to 1000 pg per tube (0.91 to 18,182 pg/ml) for androstenedione, 5α-dihydrotestosterone, testosterone, 11-deoxycortisol, cortisol, cortisone, 11-deoxycorticosterone, corticosterone, DHC, progesterone, estrone, 17β-estradiol, estriol and aldosterone. The calibration curve ranged from 2 to 1000 pg per tube (36.4 to 18,182 pg/ml) for DHEA, pregnenolone, and allopregnanolone. The on-column amount ranged from 0.041 pg to 818.2 pg per injection for all steroids (except DHEA, pregnenolone, and allopregnanolone). The on-column amount ranged from 1.64 pg to 818.2 pg per injection for DHEA, pregnenolone, and allopregnanolone. All steroids showed excellent linearity and displayed a coefficient of determination (r2) greater than 0.99 and p-values less than 0.0001 (Table 3) (Fig 2).”

While 10 μl of each standard was added to the calibration curves, these standards were then processed in the same way as the teeth samples and resuspended in 55 μl. Thus, we have revised Figure 2 to include the concentration in pg/ml instead (line 258-259).

“Calibration curve range was 2 to 1000 pg per tube (0.91 to 18,182 pg/ml) for DHEA and 0.05 to 1000 pg per tube (36.4 to 18,182 pg/ml) for cortisol and progesterone.”

How LODs and LOQs were determined is not based on any standard methods. Please recalculate, you can use the 3S/N for LODs and 10S/N method for LOQs. You need to add to your method how it was determined.

Response: The determination of the Limit of Detection (LOD) and the Lower Limit of Quantitation (LLOQ) was based on the FDA guidelines [6]. The LOD was selected as the lowest concentration at which the analyte produced a signal-to-noise (S/N) ratio ≥ 3. The LLOQ was selected as the lowest concentration with a S/N ratio ≥ 5, with an accuracy within ±20% of the nominal value and a coefficient of variation (CV) within ±20%.

We have added the LOD and LLOQ determination criteria to Table 3 (line 250-253).

“Note: The limit of detection (LOD) for aldosterone, DHC, cortisol, and corticosterone was 0.025 pg/10 μl. The LODs for progesterone, pregnenolone, and DHEA were 0.01 pg/10 μl, 0.015 pg/10 μl, and 0.2 pg/10 μl, respectively. The LOD was selected as the lowest concentration at which the analyte produced a signal-to-noise (S/N) ratio ≥ 3. The LLOQ was selected as the lowest concentration with a S/N ratio ≥ 5, with an accuracy within ±20% of the nominal value and a coefficient of variation (CV) within ±20% [51].”

References

1. Niepoth, N., Merritt, J. R., Uminski, M., Lei, E., Esquibies, V. S., Bando, I. B., Hernandez, K., Gebhardt, C., Wacker, S. A., Lutzu, S., Poudel, A., Soma, K. K., Rudolph, S., & Bendesky, A. Evolution of a novel adrenal cell type that promotes parental care. Nature. 2024;629(8014):1082-1090. doi:10.1038/s41586-024-07423-y

2. Hamden JE, Gray KM, Salehzadeh M, Kachkovski GV, Forys BJ, Ma C, Austin SH, Soma KK. Steroid profiling of glucocorticoids in microdissected mouse brain across development. Dev Neurobiol. 2021;81(2):189-206. Epub 2021/01/10. doi: 10.1002/dneu.22808. PubMed PMID: 33420760.

3. Jalabert C, Ma C, Soma KK. Profiling of systemic and brain steroids in male songbirds: Seasonal changes in neurosteroids. J Neuroendocrinol. 2021;33(1):e12922. doi:10.1111/jne.12922

4. Hamden JE, Gray KM, Salehzadeh M, Soma KK. Isoflurane stress induces region-specific glucocorticoid levels in neonatal mouse brain. J Endocrinol. 2022;255(2):61-74. Published 2022 Sep 14. doi:10.1530/JOE-22-0049

5. Salehzadeh M, Hamden JE, Li MX, Bajaj H, Wu RS, Soma KK. Glucocorticoid Production in Lymphoid Organs: Acute Effects of Lipopolysaccharide in Neonatal and Adult Mice. Endocrinology. 2022;163(2):bqab244. doi:10.1210/endocr/bqab244

6. Food and Drug Administration. Bioanalytical Method Validation Guidance for Industry. 2018. [Cited 2024 July 11]. Available from: https://www.fda.gov/drugs/guidance-compliance-regulatory-information/guidances-drugs

---

## [Decision Letter · Decision Letter 2]

13 Aug 2024

Steroid profiling in human primary teeth via liquid chromatography-tandem mass spectrometry for long-term retrospective steroid measurement

PONE-D-24-09259R2

Dear Dr. Ruolan Wu,

We’re pleased to inform you that your manuscript has been judged scientifically suitable for publication and will be formally accepted for publication once it meets all outstanding technical requirements.

Kind regards,

Tommaso Lomonaco, Ph.D

Academic Editor

PLOS ONE

Reviewers' comments:

Reviewer's Responses to Questions

**Comments to the Author**

1. If the authors have adequately addressed your comments raised in a previous round of review and you feel that this manuscript is now acceptable for publication, you may indicate that here to bypass the “Comments to the Author” section, enter your conflict of interest statement in the “Confidential to Editor” section, and submit your "Accept" recommendation.

Reviewer #1: All comments have been addressed

Reviewer #2: All comments have been addressed

2. Is the manuscript technically sound, and do the data support the conclusions?

Reviewer #1: Yes

Reviewer #2: Yes

3. Has the statistical analysis been performed appropriately and rigorously? 

Reviewer #1: Yes

Reviewer #2: N/A

4. Have the authors made all data underlying the findings in their manuscript fully available?

Reviewer #1: Yes

Reviewer #2: Yes

5. Is the manuscript presented in an intelligible fashion and written in standard English?

Reviewer #1: Yes

Reviewer #2: Yes

6. Review Comments to the Author

Reviewer #1: (No Response)

Reviewer #2: The manuscript presents a novel method for quantifying steroid levels in human primary teeth, which I believe will be of significant interest to researchers in the field. The study's subject matter is both intriguing and relevant, and the manuscript itself is well-written and easy to follow.

The authors have made commendable improvements to the Methods section. The added details have clarified the experimental procedures, making the study more comprehensible and reproducible. The inclusion of information about the method matching the calibration curve has significantly enhanced the clarity and reliability of the reported results.

I acknowledge the authors' response to the previous concern regarding the method being published elsewhere. In this manuscript, the method was initially unclear and difficult to follow, which hindered the reproducibility of the study. The authors have now addressed this issue effectively by properly detailing the method, thereby providing a complete and understandable description.

Overall, the revised manuscript represents a valuable contribution to the literature, and I support its publication with the improved clarity and thoroughness of the Methods section.

7. PLOS authors have the option to publish the peer review history of their article (what does this mean?). If published, this will include your full peer review and any attached files.

Reviewer #1: No

Reviewer #2: No

---

## [Editor Report · Acceptance letter]

19 Aug 2024

PONE-D-24-09259R2 

PLOS ONE

Dear Dr. Wu, 

I'm pleased to inform you that your manuscript has been deemed suitable for publication in PLOS ONE. Congratulations! Your manuscript is now being handed over to our production team.

Kind regards, 

on behalf of

Dr. Tommaso Lomonaco 

Academic Editor

PLOS ONE